# An Improved Projector Calibration Method by Phase Mapping Based on Fringe Projection Profilometry

**DOI:** 10.3390/s23031142

**Published:** 2023-01-19

**Authors:** Yabin Liu, Bingwei Zhang, Xuewu Yuan, Junyi Lin, Kaiyong Jiang

**Affiliations:** 1Fujian Key Laboratory of Special Energy Manufacturing, Huaqiao University, Xiamen 361021, China; 2Xiamen Key Laboratory of Digital Vision Measurement, Huaqiao University, Xiamen 361021, China

**Keywords:** fringe projection profilometry, projector calibration, local random sample consensus, cubic polynomial fitting, phase mapping

## Abstract

Aiming at the problem of the low accuracy of projector calibration in a structured light system, an improved projector calibration method is proposed in this paper. One of the key ideas is to estimate the sub-pixel coordinates in the projector image plane using local random sample consensus (RANSAC). A bundle adjustment (BA) algorithm is adopted to optimize the calibration parameters to further improve the accuracy and robustness of the projector calibration. After system calibration and epipolar rectification, the mapping relationship between the pixel coordinates and the absolute phase in the projector image plane is established by using cubic polynomial fitting, and the disparity is rapidly solved by using the mapping relationship, which not only ensures the measurement accuracy, but also improves the measurement efficiency. The experimental results demonstrated that the average re-projection error after optimization is reduced to 0.03 pixels, and the proposed method is suitable for high-speed 3D reconstruction without the time-consuming homogenous point searching.

## 1. Introduction

Fringe projection profilometry (FPP) is a spatial coding technology that uses a projector to project a series of sinusoidal fringe patterns onto the surface of the measured object. A camera captures the deformed fringe patterns modulated by the surface of the measured object, and then reconstructs the 3D morphology of the measured object [1]. It is widely used in many fields, including object detection, positioning and 3D measurement [2,3]. Phase-height mapping (PHM) [4] and stereo vision (SV) method [5] are two typical approaches for FPP to complete 3D reconstruction. However, the PHM method generally requires a reference plane to compute the difference between the phases on the reference plane and surface of the measured object. The height is calculated from this phase difference. A precise device is needed to ensure the accurate translation pose for establishment of the PHM model through calibration, which may increase the complexity and cost. The SV method considers the influence of lens distortion, which can obtain a higher accuracy of 3D reconstruction, and it is more flexible than PHM method in calibration. The SV system of two cameras can be directly calibrated by acquiring images of left and right cameras, which has been quite mature [6,7]. In order to improve the speed, we intend to use a camera and a projector to form a SV system to avoid time-consuming homogenous points searching. However, the projector cannot directly capture images of the chessboard; the method to achieve high-accuracy projector calibration still faces many challenges.

There are two principal methods for projector calibration: One is to project feature points onto a calibration board, and then calculate the homography matrix by the relationship of feature points coordinates between the projector and camera. By doing so, the projector is calibrated in the same way as a camera [8,9]. The other is based on phase mapping technology. The projector projects a sinusoidal fringe pattern on the calibration board, and the images on the calibration board are captured by a camera. The phases of the feature points on the calibration board can be calculated, so that the coordinates of feature points in camera image can be converted to the projector according to the equal phase value, and the further calibration work is carried out [10,11]. The former kind of projector calibration method will cause the propagation of camera calibration error. The later can avoid the propagation of camera calibration error. However, when using a chessboard for calibration, the sinusoidal fringe will lose its sinusoidal characteristics due to the obviously different reflectivity of the chessboard in the black and white areas. That is, the noise of background causes low signal-to-noise ratio (SNR), resulting in lower accuracy of chessboard corners extraction [12]. To solve this issue, Wilm J et al. [13] used a light-colored chessboard to reduce the reflectivity change on the edge of chessboard squares, but the extraction of chessboard corners is relatively difficult. Li et al. [14] used the circle array calibration board to interpolate the phase of the pixels around the center of the circle and obtain the phase for calibration, which improved the robustness of the projector calibration. However, eccentric error and edge deformation in the circular mark points will produce additional error. In another aspect, due to the error caused by various factors, BA algorithm is often introduced in vision calibration to reduce the re-projection error and optimize the calibration parameters. The use of BA algorithm can achieve the jointly optimal estimate of the system parameters and 3D world coordinates of the feature points by minimizing the model of the error function [15,16].

In the process of 3D reconstruction, the SV method requires a considerable amount of time to find the corresponding feature points, which greatly affects the efficiency of 3D reconstruction. Cai et al. [17] proposed a fast 3D reconstruction method based on the phase-3D mapping lookup table of the back-projection SV model. By formulating a two-step calibration strategy of ray re-projection calibration and sampling-mapping calibration, a mapping lookup table from phase to 3D coordinates of the measured points was established. However, this method needs additional sampling calibration to establish the relationship between phase and 3D coordinate, which cannot guarantee the measurement accuracy outside the sampling range, and the calibration results occupy a lot of system memory, thus increasing the complexity and difficulty of practical application.

Based on the above methods, this paper proposes an improved projector calibration method based on FPP by phase mapping, using a cost-effective black/white chessboard. First, by using local RANSAC, the problem of low SNR of the chessboard is solved, thus obtaining the sub-pixel coordinates of the projector. Second, the BA algorithm is utilized to diminishing the printing error of mark points, further improving the calibration accuracy. In addition, we establish the mapping relationship between the absolute phase and the projector coordinates by the cubic polynomial for more flexible rapid 3D reconstruction. The calibration and measurement experiments verify the effectiveness of the method.

## 2. Camera and Projector Model of the Measurement System

The 3D measurement system mentioned in this experiment is composed of a projector and a camera. Generally speaking, we regard the projector as an inverse camera and calibrate the camera and projector by using the pinhole imaging model and lens distortion model [18,19]. Calibrating a camera or projector is to solve its intrinsic and extrinsic parameters. The intrinsic parameters mainly include focal length, principal point coordinates and distortion parameters, while the extrinsic parameters describe the relationship between the image coordinate systems of cameras or projector and the world coordinate system, including a rotation matrix and a translation vector.

Figure 1 shows the structure of the measurement system, where Q is a corner on the calibration board, and the coordinate in the world coordinate system is $\left(X_{w},Y_{w},Z_{w}\right)$. Q produces its images at the points $q_{c}$ and $q_{p}$ in the image plane of the camera and projector, respectively. Based on the pinhole imaging model, the relationship between the camera pixel coordinates and world coordinates can be described in the following form:(1)$$Z_{c}\left[\begin{array}{c} u_{c}\\ v_{c}\\ 1 \end{array}\right]=A_{c}\left[\begin{array}{c} X_{c}\\ Y_{c}\\ Z_{c} \end{array}\right]=\left[\begin{array}{ccc} f_{c}/du_{c} & 0 & u_{c0}\\ 0 & f_{c}/dv_{c} & v_{c0}\\ 0 & 0 & 1 \end{array}\right]\left[\begin{array}{c} X_{c}\\ Y_{c}\\ Z_{c} \end{array}\right]=A_{c}[R_{c},T_{c}]\left[\begin{array}{c} X_{w}\\ Y_{w}\\ Z_{w}\\ 1 \end{array}\right]$$

Similarly, in the projector:(2)$$Z_{p}\left[\begin{array}{c} u_{p}\\ v_{p}\\ 1 \end{array}\right]=A_{p}\left[\begin{array}{c} X_{p}\\ Y_{p}\\ Z_{p} \end{array}\right]=\left[\begin{array}{ccc} f_{p}/du_{p} & 0 & u_{p0}\\ 0 & f_{p}/dv_{p} & v_{p0}\\ 0 & 0 & 1 \end{array}\right]\left[\begin{array}{c} X_{p}\\ Y_{p}\\ Z_{p} \end{array}\right]=A_{p}[R_{p},T_{p}]\left[\begin{array}{c} X_{w}\\ Y_{w}\\ Z_{w}\\ 1 \end{array}\right]$$
where $f_{c}$, $f_{p}$ and $du_{c}$, $dv_{c}$, $du_{p}$, $dv_{p}$ are known as the focal length and pixel size in different directions of the camera and projector, respectively. $\left(u_{c0},v_{c0}\right)$ and $\left(u_{p0},v_{p0}\right)$ represent the centers of the pixel coordinates of the camera and projector, respectively. $A_{c}$ and $A_{p}$ are the intrinsic parameter matrices of the camera and projector, respectively. $\left[R_{c},T_{c}\right]$ and $\left[R_{p},T_{p}\right]$ represent the extrinsic parameter matrices of the camera and projector, respectively, which consist of a 3 × 3 rotation matrix and a 3 × 1 translation vector, respectively.

In addition, due to the distortion of the camera and projector lens, there is a deviation between the ideal pixel coordinates (x,y) and the actual pixel coordinates $\left(\hat{x},\hat{y}\right)$. The relationship between (x,y) and $\left(\hat{x},\hat{y}\right)$ can be described by the Brown–Conrady lens distortion model:(3)$$\left[\begin{array}{c} \hat{x}\\ \hat{y} \end{array}\right]=\left[\begin{array}{c} x\\ y \end{array}\right]+\sum_{n=1,2}k_{n}r^{2n}\left[\begin{array}{c} x\\ y \end{array}\right]+\left[\begin{array}{c} 2p_{1}xy+p_{2}(r^{2}+2x^{2})\\ 2p_{2}xy+p_{1}(r^{2}+2y^{2}) \end{array}\right]$$
where $k=[k_{1},k_{2}]$ and $p=[p_{1},p_{2}]$ represent the radial and tangential distortion coefficients of the lens, respectively, and $r=\sqrt{x^{2}+y^{2}}$ is the distance from the distorted image point to the principal point.

Generally, the camera can extract the feature points by capturing the images of the calibration board at different positions to complete the calibration. Unlike the camera, the projector cannot capture the feature points. We established the mapping relationship between the camera and projector by projecting phase-shifting fringe patterns to convert the coordinates of feature points from camera to projector. This method is detailed in Section 3.

## 3. High-Accuracy Projector Calibration Method

### 3.1. Phase Calculation

In a structured light system, the phase-shifting method can establish the correspondence between the camera and projector pixel coordinates, which plays a vital role in system calibration. This section describes how to determine the camera and projector pixel coordinates for the feature points, i.e., corners on the chessboard. As shown in Figure 1, we denoted the pixel coordinates of chessboard corners captured by the camera as $\left(u_{c},v_{c}\right)$, and denoted the corresponding pixel coordinates of the projector as $\left(u_{p},v_{p}\right)$. One of the difficulties in projector calibration is how to determine the correspondence between the pixel coordinates of the projector and feature points on the calibration board. In this paper, we solved the corresponding problem using phase mapping technology. Firstly, the sinusoidal fringe in horizontal and vertical directions generated by the computer can be described as:(4)$$\left\{\begin{array}{l} I_{i}^{u}(u_{p},v_{p})=\alpha +\beta \cos[2\pi u_{p}/\lambda_{u}+\varphi_{i}]\\ I_{i}^{v}(u_{p},v_{p})=\alpha +\beta \cos[2\pi v_{p}/\lambda_{v}+\varphi_{i}] \end{array}\right.$$
where $I_{i}(u_{p},v_{p})$ is the light intensity at a projector pixel $\left(u_{p},v_{p}\right)$. α and β are the background and modulation intensity, respectively, and λ is the fringe period. When the generated sinusoidal fringes are projected on the measured object by the projector, the fringe is distorted on the surface of the object, forming a deformed fringe. The light intensity of the deformed fringe images can be described as:(5)$$\left\{\begin{array}{l} {\tilde{I}}_{i}^{u}(u_{c},v_{c})=a(u_{c},v_{c})+b(u_{c},v_{c})\cos[\Phi_{u}(u_{c},v_{c})+\varphi_{i}]\\ {\tilde{I}}_{i}^{v}(u_{c},v_{c})=a(u_{c},v_{c})+b(u_{c},v_{c})\cos[\Phi_{v}(u_{c},v_{c})+\varphi_{i}] \end{array}\right.$$
where ${\tilde{I}}_{i}(u_{c},v_{c})$ is the light intensity at $\left(u_{c},v_{c}\right)$ on the captured images. $a(u_{c},v_{c})$ and $b(u_{c},v_{c})$ are the background intensity and the modulation intensity at a camera pixel $\left(u_{c},v_{c}\right)$, respectively, and $\varphi_{i}$ is the phase-shifting value in ith step. In this paper, we adopted the three-step phase-shifting method to generate three sinusoidal fringe images with the same phase-shifting interval of 2π/3, and the wrapped phase can be obtained as follows:(6)$$\left\{\begin{array}{l} \varphi_{u}(u_{c},v_{c})=\arctan\left\{\frac{\sqrt{3}\left[{\tilde{I}}_{1}^{u}(u_{c},v_{c})-{\tilde{I}}_{3}^{u}(u_{c},v_{c})\right]}{2{\tilde{I}}_{2}^{u}(u_{c},v_{c})-{\tilde{I}}_{1}^{u}(u_{c},v_{c})-{\tilde{I}}_{3}^{u}(u_{c},v_{c})}\right\}\\ \varphi_{v}(u_{c},v_{c})=\arctan\left\{\frac{\sqrt{3}\left[{\tilde{I}}_{1}^{v}(u_{c},v_{c})-{\tilde{I}}_{3}^{v}(u_{c},v_{c})\right]}{2{\tilde{I}}_{2}^{v}(u_{c},v_{c})-{\tilde{I}}_{1}^{v}(u_{c},v_{c})-{\tilde{I}}_{3}^{v}(u_{c},v_{c})}\right\} \end{array}\right.$$
where $\varphi_{u}(u_{c},v_{c})$ and $\varphi_{v}(u_{c},v_{c})$ are the wrapped phase in horizontal and vertical directions, respectively. The phase is periodically truncated at [−π,π] because of the use of arctangent function in Equation (6). In this paper, the multi-frequency phase unwrapping method [20] was utilized to procure the absolute phases $\Phi_{u}(u_{c},v_{c})$ and $\Phi_{v}(u_{c},v_{c})$. The corresponding projector pixel coordinates can be calculated by the absolute phase on chessboard corners from Equation (7), and the relationship between the coordinates of projector and chessboard is established. The projector can serve as a camera, and the calibration process is carried out in the same way as a camera.
(7)$$\left\{\begin{array}{l} u_{p}=\frac{\lambda_{u}\Phi_{u}(u_{c},v_{c})}{2\pi }\\ v_{p}=\frac{\lambda_{v}\Phi_{v}(u_{c},v_{c})}{2\pi } \end{array}\right.$$

### 3.2. Sub-Pixel Coordinate Extraction Based on the Local RANSAC

A black/white chessboard with the corners as feature points was used for projector calibration. In the process of phase extracting, due to the low-pass property of the camera lens and the low reflectivity of the black area on the chessboard, the SNR is low. By setting an appropriate sampling threshold value of modulation intensity, the areas with modulation intensity below the threshold are filtered out and the other areas near the corners can be retained and used for the high-quality phase calculation. The threshold was set to 10 in our system, and the modulation intensity $b(u_{c},v_{c})$ is calculated by Equation (8):(8)$$b(u_{c},v_{c})=\frac{\sqrt{3{\left[{\tilde{I}}_{1}(u_{c},v_{c})-{\tilde{I}}_{3}(u_{c},v_{c})\right]}^{2}+{\left[2{\tilde{I}}_{2}(u_{c},v_{c})-{\tilde{I}}_{1}(u_{c},v_{c})-{\tilde{I}}_{3}(u_{c},v_{c})\right]}^{2}}}{3}$$

However, since the step change of the reflectivity occurs on the edges of the black and white areas of the chessboard, it causes serious random phase errors. We used the local RANSAC [21] to effectively suppress the noise influence on extracted phases at the corners of selected areas.

As shown in Figure 2, a square area with a side length of 50 pixels on each corner of the chessboard was taken as the fitting area, and the fitting plane was fitted with 10 points randomly selected within the square area. This process was carried out 50 times, and the optimal fitting plane was determined from 50 fitting results with the minimum fitting error.

After local plane fitting, the phase of corners in sub-pixel of the chessboard can be obtained through interpolation on the fitting plane. Accordingly, by using mapping function described in Equation (7), we can obtain the sub-pixel corners in projector. To verify the effectiveness of the algorithm, we adopted different interpolation methods to extract the phases of chessboard corners, and the phase error is shown in Figure 3. Figure 3a shows the errors of the sub-pixel corners extracted by bilinear interpolation. Figure 3b shows the errors of the sub-pixel corners extracted by fitting the phase directly, and Figure 3c shows the errors of the sub-pixel corners extracted by fitting the phase of the local white squares with RANSAC. The experimental results show that the coordinates of chessboard corners extracted by the local RANSAC are less influenced by noise.

### 3.3. Optimized Calibration by BA Algorithm

To further improve the accuracy of projector calibration, the BA algorithm was introduced into the proposed method in the step of global optimization [22]. To obtain the globally optimal estimation of the parameters, we defined the following objective function E:(9)$$\begin{array}{l} E=\min\sum_{i=1}^{N}\sum_{j=1}^{M}\left\{||m_{cij}-{\hat{m}}_{cij}(A_{c},K_{c},P_{c},M_{ci},X_{j})|{|}^{2}\right.\\ \left.+||m_{pij}-{\hat{m}}_{pij}(A_{p},K_{p},P_{p},M_{ci},M_{i},X_{j})|{|}^{2}\right\} \end{array}$$
where $A_{c}$ and $A_{p}$ are known as the intrinsic parameter matrix of the camera and projector, respectively; and $K_{c}$, $K_{p}$ and $P_{c}$, $P_{p}$ as radial distortion coefficient and tangential distortion coefficient of the camera and projector, respectively. The subscript i and j respectively denote the *i*th pose and the *j*th corner of the chessboard, respectively. $M_{ci}$ includes the rotation matrix $R_{c}$ and translation vector $T_{c}$ from world coordinates to camera coordinates in the *i*th pose. $M_{i}$ includes the rotation matrix R and translation vector T from the projector coordinates to camera coordinates in the *i*th pose. $X_{j}$ represents the 3D coordinates of feature points on the chessboard. $m_{cij}$ and $m_{pij}$ represent the coordinates of feature points in the image plane of the camera and projector, respectively. ${\hat{m}}_{cij}$ and ${\hat{m}}_{pij}$ represent the coordinates of feature points re-projected to the image plane of the camera and projector through the pinhole imaging model, respectively.

The calibration process can be summarized in the following steps:

(1) The camera is used to capture the images before and after projecting the structured light onto the chessboard. Then, the image coordinates of the chessboard corners from the camera image are extracted and the corresponding phases are calculated through 3-step phase-shifting. The corresponding corner coordinates on the projector image are obtained through the phase-mapping process.

(2) According to the above image coordinates, Zhang’s calibration method [6] is utilized to calibrate the projector and camera. Using the intrinsic and extrinsic parameters, the 3D coordinates are re-projected on the images of camera and projector, respectively.

(3) The objective function of the BA algorithm is established according to the re-projection error, and then the system parameters and the world coordinates of the checkerboard corners are globally adjusted by the Levenberg–Marquardt (LM) algorithm [23] until the objective function converges to the minimum.

### 3.4. Three-Dimensional Reconstruction Based on Phase Mapping

After epipolar rectification [24], it only needs to search the matching points of the left and right images on a line. However, for high-speed 3D measurement, especially in large-scale 3D reconstruction, the traversal search process is also time-consuming even after the optimization of epipolar rectification.

The correspondence between the phase $\phi_{p}$ and the coordinate $u_{p}$ on the coding fringes image of projector is known. The mapping points of the camera and projector images should satisfy $\phi_{p}(u_{p}^{*},v_{p}^{*})=\phi_{c}(u_{c}^{*},v_{c}^{*})$ and $v_{c}^{*}=v_{p}^{*}$. By the unique correspondence between the absolute phase and the image coordinates, the absolute phase $\phi_{c}$ of the camera can be mapped to the image coordinate $u_{p}$ of the projector. According to this relationship, the disparity of the corresponding pixels can be expressed as:(10)$$\left\{\begin{array}{l} Disparity=f\left[\phi_{p}(u_{p}^{*},v_{p}^{*}),v_{p}^{*}\right]-u_{c}^{*}\\ \phi_{p}(u_{p}^{*},v_{p}^{*})=\phi_{c}(u_{c}^{*},v_{c}^{*})\\ v_{p}^{*}=v_{c}^{*} \end{array}\right.$$
where $\left(u_{p}^{*},v_{p}^{*}\right)$ and $\left(u_{c}^{*},v_{c}^{*}\right)$ represent the projector and camera pixel coordinates after epipolar rectification, respectively. The function f() represents the correspondence between the absolute phase and the pixel coordinates of the projector fitted by a polynomial function. Based on $\phi_{c}=\phi_{p}$ and $v_{c}^{*}=v_{p}^{*}$, we can obtain $f\left[\phi_{p}(u_{p}^{*},v_{p}^{*}),v_{p}^{*}\right]=f\left[\phi_{c},v_{c}^{*}\right]$ from Equation (10). Then, we can directly map the phase to the corresponding disparity without homologous point searching, which further improves the measurement speed.

The polynomial function f() was used for fitting the correspondence between the absolute phases and the pixel coordinates of the projector. After the nonlinear lens distortion correction and the image rectification, the correspondence changed from linear to nonlinear. In order to improve the measurement accuracy, the correspondence needs to be fitted by high-order polynomial. To explore the influence of fitting accuracy on the accuracy of 3D reconstruction, the mapping relationship between the phase and the projector pixel coordinates was established by using different fitting methods. Table 1 shows the fitting results of polynomial functions with different methods. The root-mean-square error (RMSE) of polynomial fitting and the 3D reconstruction of a chessboard are provided. The results show that the full-field RMSE of using cubic polynomial fitting is 0.2185 pixels, which is reduced by 99.77% and 71.95%, respectively, compared with the methods of linear and quadratic polynomial fitting. The results of the 3D reconstruction data of the chessboard with different methods show that the full-field RMSE of using cubic polynomial fitting is 0.1209 mm, which is reduced by 93.2% and 22.2%, respectively, compared with the methods of linear and quadratic polynomial fitting. The influence of different fitting methods on the reconstruction accuracy can be seen in Figure 4a–c. To obtain better results in practice, we used the cubic polynomial to fit. The overall flow chart of proposed method is demonstrated in Figure 5.

## 4. Experiment and Discussion

### 4.1. System Setup

To verify the effectiveness and correctness of the proposed method, an experimental system with FPP technology was built, as shown in Figure 6. The system consists of a computer, a digital projector (ElecShark ES3000T), and a CCD camera (DMK23U274). The camera and projector were fixed on an assembled frame. The image acquisition and system calibration processes were performed on the software developed by Microsoft Visual Studio 2013.

### 4.2. Calibration Experiment

A black/white chessboard of 12 × 9 squares was used as the calibration board, with each square having a size of 10 × 10 mm^2^. During the calibration, the system was calibrated with 15 random poses uniformly filled in the measuring volume.

After calibration, the 3D reconstructed coordinates of the corners on the chessboard in various poses were re-projected to the camera and projector image planes, respectively, and then compared with the coordinates of corners used in calibration to figure out the re-projection error. The re-projection error of the projector and camera are shown in Figure 7a–c. Then, the calibration accuracy of the camera and projector was further improved by introducing the BA algorithm, and the re-projection error of the camera and projector after BA algorithm are shown in Figure 7d–f, where points in different colors mean the re-projected error of each corner of chessboard in different poses.

Figure 7 shows the re-projection error distribution of the camera and projector before and after the BA algorithm. Based on the results of the re-projection error distribution given in Figure 7c,f, the overall average error without the BA algorithm is 0.11 pixels. After optimization with the BA algorithm, the overall average error is reduced to 0.03 pixels. Figure 8 shows the results of the calibrated intrinsic and extrinsic parameters, and the calibration diagram. The reprojection errors of the camera, projector, and both (the overall mean error) are calculated and compared with different methods in Table 2. Therefore, the calibration accuracy can be significantly improved by introducing the BA algorithm.

### 4.3. Measurement Experiment

To verify the accuracy of the proposed method, we used the calibrated measurement system to measure the standard ball plate. As shown in Figure 9a,b, the diameters of A ball and B ball are 38.0940 mm and 38.0887 mm, respectively, and the distance between the two balls is 100.0870 mm. The experimental results are shown in Figure 9c.

We used the calibrated measurement system to measure the standard ball plate. After the measurement, we can obtain the diameter of the standard balls, the distance between the ball centers, and the error of these data, as shown in Figure 9c. The measured diameter of ball A is DA = 38.1056 mm, the measured diameter of ball B is DB = 38.0671 mm, and the measured distance between the two balls is 100.212 mm. Therefore, the proposed method can achieve 3D reconstruction with high accuracy.

In addition, to evaluate the speed of 3D measurement, a time-consuming comparison was carried out between our proposed method and a typical SV method in the measurement of three different objects, as shown in Figure 10, namely, a stepped metal block (a), a water sprinkler part (b), and a metal plate (c). The experimental results are shown in Figure 10d–f and Table 3. In general, the SV method needs to take more time to find the matching points between left and right camera images by a time-consuming searching process. Our method finds the matching points simply by using a mapping relationship between the pixel coordinates and the absolute phase in the projector image plane without searching process, so that it can quicken the 3D reconstruction. From Table 3, we can observe that our method has a higher reconstruction speed.

## 5. Conclusions

Aiming at the problem of low calibration accuracy of the projector in FPP system, the proposed method studied the projector calibration method based on phase-mapping technology. By using a local RANSAC algorithm, the problem of low SNR of the chessboard was solved. After that, the use of BA algorithm further improved the calibration accuracy by diminishing the error of the system parameters and the 3D feature points simultaneously. In addition, in 3D reconstruction, the disparity can be quickly generated according to the mapping relationship between the absolute phase and pixel coordinates of the projector after epipolar rectification, thereby achieving the high-speed 3D measurement. Experimental results demonstrate the accuracy of the proposed method, and the proposed method is suitable for high-speed 3D reconstruction without time-consuming homogenous point searching.

## Figures and Tables

**Figure 1 sensors-23-01142-f001:**
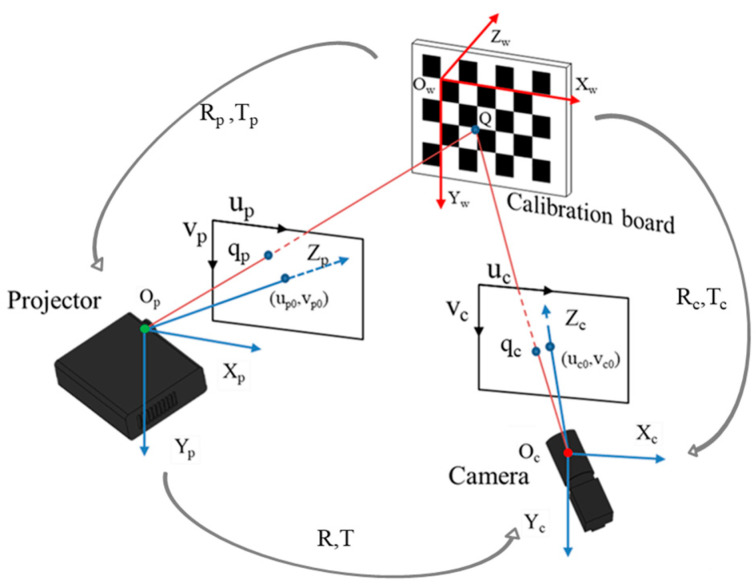
The structure of the measurement system.

**Figure 2 sensors-23-01142-f002:**
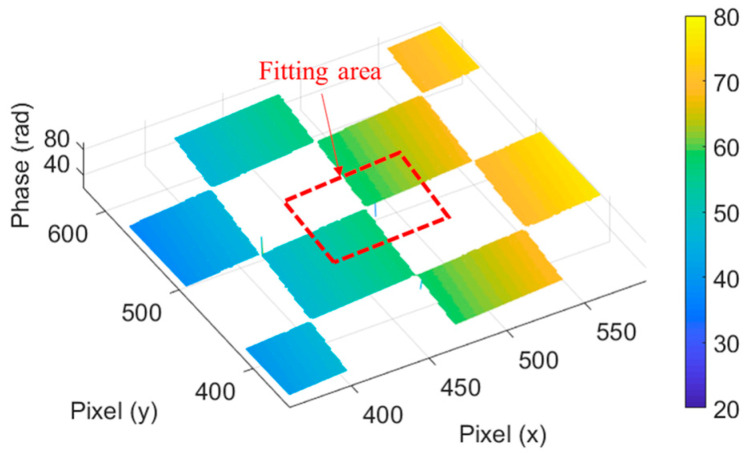
The fitting area at chessboard corners.

**Figure 3 sensors-23-01142-f003:**
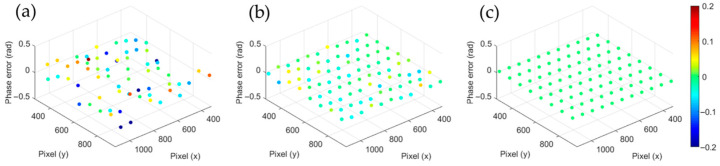
Experiment of phase error suppression at checkerboard corners. (**a**) The phase error of sub-pixel corners extracted by bilinear interpolation, (**b**) by fitting the phase of the local white squares directly, and (**c**) by fitting the phase of the local white squares with RANSAC.

**Figure 4 sensors-23-01142-f004:**
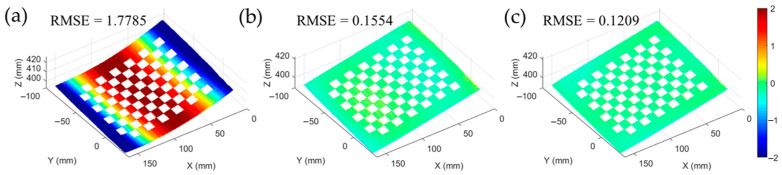
The 3D reconstruction of a checkerboard: (**a**) by using linear polynomial fitting method, (**b**) by using quadratic polynomial fitting method, and (**c**) by using cubic polynomial fitting method.

**Figure 5 sensors-23-01142-f005:**
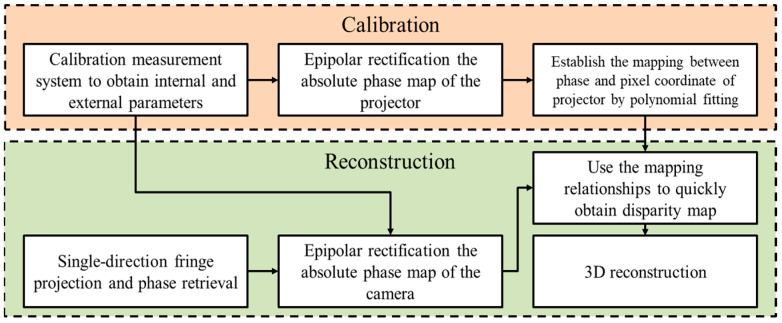
The overall flow chart of the proposed method.

**Figure 6 sensors-23-01142-f006:**
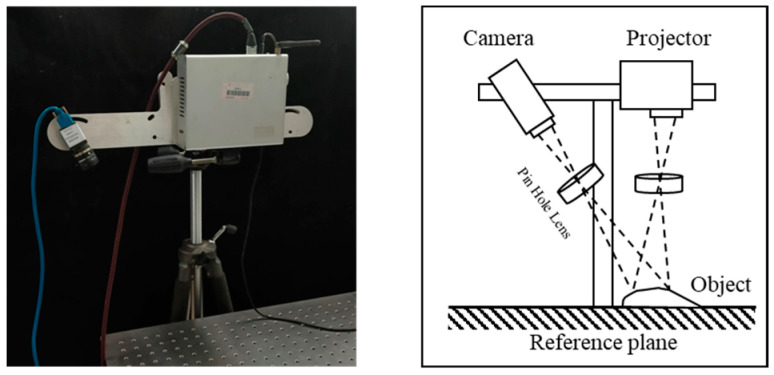
The experimental FPP system.

**Figure 7 sensors-23-01142-f007:**
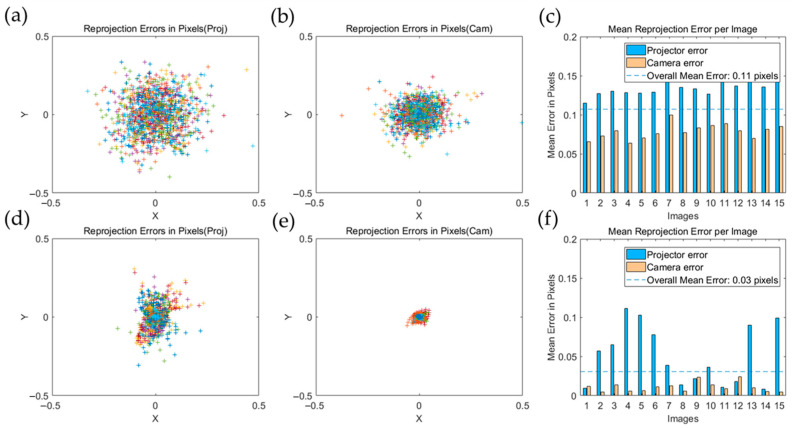
The re-projection error: (**a**) of a projector before the BA algorithm, (**b**) of the camera before the BA algorithm, (**c**) of 15 random poses before the BA algorithm, (**d**) of the projector after the BA algorithm, (**e**) of the camera after the BA algorithm, and (**f**) of 15 random poses after the BA algorithm.

**Figure 8 sensors-23-01142-f008:**
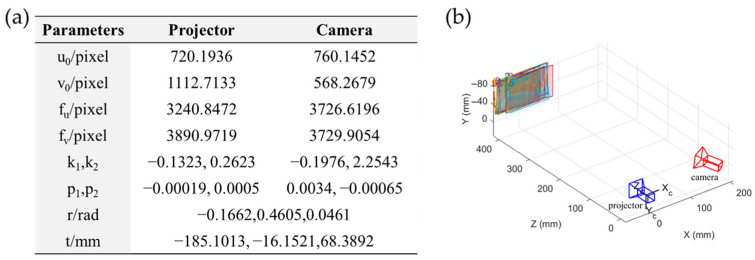
Calibration results of an FPP system using the proposed method. (**a**) The intrinsic and structural parameters of the FPP system and (**b**) the diagram of stereo model of the FPP system.

**Figure 9 sensors-23-01142-f009:**
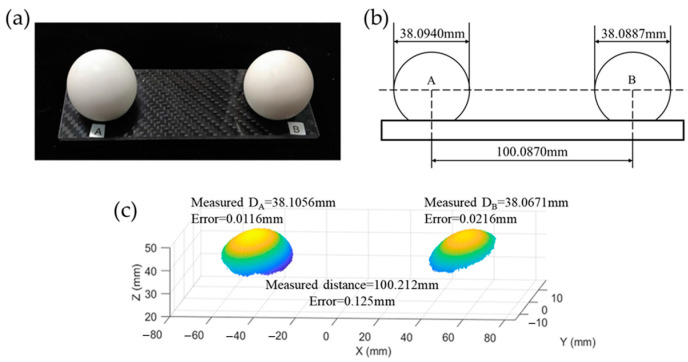
Reconstruction results of the standard ball plate: (**a**) physical map, (**b**) schematic diagram of size, and (**c**) 3D measurement result.

**Figure 10 sensors-23-01142-f010:**
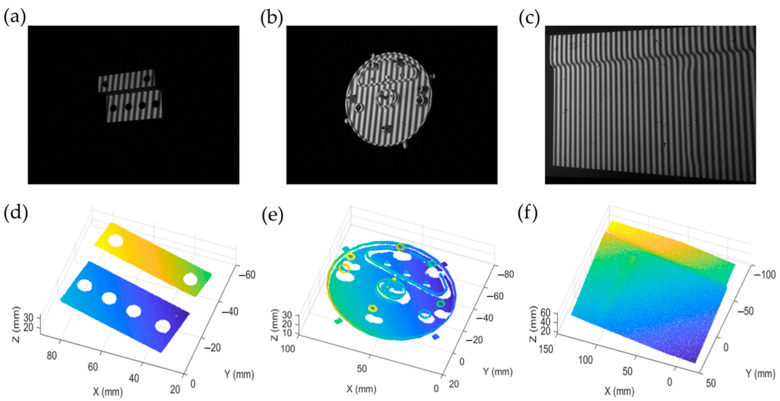
The captured fringe images of different objects: (**a**) a stepped metal block, (**b**) a water sprinkler part, (**c**) a metal plate, (**d**) the measurement result of (**a**), (**e**) the measurement result of (**b**), and (**f**) the measurement result of (**c**).

**Table 1 sensors-23-01142-t001:** Polynomial fitting results with different fitting methods.

Fitting Method	Polynomial Fitting Results of *f*()	RMSE	
		Polynomial Fitting	3D Reconstruction
Linear polynomial	$u_{p}=-234+0.06224v_{p}+8.49\phi_{p}$	9.753 pixel	1.7785 mm
Quadratic polynomial	$\begin{array}{l} u_{p}=-190.6+0.04976v_{p}+7.596\phi_{p}\\ +0.000111v_{p}\phi_{p}+0.003667\phi_{p}{}^{2} \end{array}$	0.7789 pixel	0.1554 mm
Cubic polynomial	$\begin{array}{l} u_{p}=-196.3+0.05106v_{p}+7.807\phi_{p}\\ +8.376e^{-5}v_{p}\phi_{p}+0.001583\phi_{p}{}^{2}\\ +1.22e^{-7}v_{p}\phi_{p}{}^{2}+5.95e^{-6}\phi_{p}{}^{3} \end{array}$	0.2185 pixel	0.1209 mm

**Table 2 sensors-23-01142-t002:** Calibration reprojection errors (pixels) with different methods.

Method	Reprojection Errors (Pixel)		
	Camera	Projector	Overall Mean Error
Moreno and Taubin [18]	0.15	2.58	1.83
Global homography	0.15	7.45	5.27
Huang’s method [25]	0.26	0.17	0.22
Proposed	0.02	0.06	0.03

**Table 3 sensors-23-01142-t003:** Data related to the efficiency of 3D reconstruction.

Scenes	Proposed Method		SV Method	
	Number of Points	Time Cost (s)	Number of Points	Time Cost (s)
Metal block	145,293	0.207	142,587	0.76
Water sprinkler	378,727	0.235	360,769	1.99
Metal plate	1,543,808	0.412	1,617,944	5.08

## Data Availability

Data are contained within the article.

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
