# Peer review of "An Improved Projector Calibration Method by Phase Mapping Based on Fringe Projection Profilometry"

_sensors, 2023, doi:10.3390/s23031142_

Round 1

Reviewer 1 Report

This paper presents a method to calibrate Fringe-projection-system in stereo-vision principle. It used a black-white checkerboard as the calibration rig to calibrate the camera, and by mapping the projector pixels with the phase distributions to calibrate the projector, finally to refine the systematic parameters by BA algorithm. Overall, the article is well organized and its presentation is good. However, some issues still need to be improved:

(1)  The authors failed to list the manufacturing accuracy of the checkerboard, and it is not clear whether the world coordinates of the checkerboard corners will be refined in the LM algorithm.

(2) The authors only provide the results by the proposed calibration method, however, as the authors listed, the same system can also be calibrated by PHM. Therefore, it will be much better to quantitatively compare PHM and the proposed method.

Author Response

Response letter

Dear Editors and Reviewers:

Thank you for your letter and the comments from the reviewers on our paper “An improved projector calibration method by phase mapping based on fringe projection profilometry” (Manuscript ID: sensors-2119358). Those comments are all quite insightful and helpful. We would like to thank the reviewer for reviewing our manuscript and helping us improve it. The following are the primary corrections in the manuscript and replies to the reviewer’s comments:

Responds to the reviewer’s comments:

Reviewer #1:

This paper presents a method to calibrate Fringe-projection-system in stereo-vision principle. It used a black-white checkerboard as the calibration rig to calibrate the camera, and by mapping the projector pixels with the phase distributions to calibrate the projector, finally to refine the systematic parameters by BA algorithm. Overall, the article is well organized and its presentation is good. However, some issues still need to be improved:

Question 1:

The authors failed to list the manufacturing accuracy of the checkerboard. Line 217: It is not clear whether the world coordinates of the checkerboard corners will be refined in the LM algorithm.

Response 1:

Thanks for the reviewer's reminding. First, since we used a printed checkerboard in our experiment, its manufacturing accuracy is not listed in the manuscript. As the reviewer said, the world coordinates of the checkerboard corners will be refined in the LM algorithm, and we will make this clear in the revised manuscript.

We change the sentence “and then the system parameters and 3D coordinates of the feature points are globally adjusted by the Levenberg-Marquardt (LM) algorithm” to “and then the system parameters and the world coordinates of the checkerboard corners are globally adjusted by the Levenberg-Marquardt (LM) algorithm” in revised manuscript (Line 218-219).

We change the sentence “Second, the BA algorithm is utilized to optimize the system parameters, further improving the calibration accuracy.” to “Second, the BA algorithm is utilized to diminishing the printing error of mark points, further improving the calibration accuracy.” in revised manuscript (Line 80-81).

Question 2:

Line 15: The authors only provide the results by the proposed calibration method, however, as the authors listed, the same system can also be calibrated by PHM. Therefore, it will be much better to quantitatively compare PHM and the proposed method.

Response 2:

Thanks to the reviewer’s comment. Our method in this manuscript is based on SV method, and the calibration process is different from the PHM method. The method of PHM requires a precise mobile platform for calibration, so it is inappropriate to compare the PHM method with the method in this manuscript. We added a comparison of the re-projection errors with different methods, as shown in Table 2 in revised manuscript (Line 300). And the reconstruction speed is compared with SV method, as shown in Table 3 in revised manuscript (Line 329).

Reviewer 2 Report

The authors proposed a method for pro-cam system calibration based on fringe projection profilometry. I believe the novelty of the manuscript is limited. Phase shifting method (as well as gray code) are frequently used for correspondence extraction in the literature. Moreover, I have some other major concerns regarding the manuscript as follows:

-line 78: by using local RANSAC, the sub-pixel coordinates of the projector with high accuracy can be obtained. RANSAC is just an outlier removal algorithm. How it can increase the accuracy?

-line 80: In addition, instead of a lookup table, the mapping relationship between the absolute phase and the projector coordinates by the cubic polynomial can become more flexible for rapid 3D reconstruction. Actually, look up tables are tools to replace runtime computation with a simple array indexing. How using a fitting process instead of LUT makes the algorithm more flexible?

-The proposed method should be compared to some baseline algorithms.

-The English should be improved

Author Response

Response letter

Dear Editors and Reviewers:

Thank you for your letter and the comments from the reviewers on our paper “An improved projector calibration method by phase mapping based on fringe projection profilometry” (Manuscript ID: sensors-2119358). Those comments are all quite insightful and helpful. We would like to thank the reviewer for reviewing our manuscript and helping us improve it. The following are the primary corrections in the manuscript and replies to the reviewer’s comments:

Responds to the reviewer’s comments:

Reviewer #2:

The authors proposed a method for pro-cam system calibration based on fringe projection profilometry. I believe the novelty of the manuscript is limited. Phase shifting method (as well as gray code) are frequently used for correspondence extraction in the literature. Moreover, I have some other major concerns regarding the manuscript as follows:

Question 1:

Line 78: by using local RANSAC, the sub-pixel coordinates of the projector with high accuracy can be obtained. RANSAC is just an outlier removal algorithm. How it can increase the accuracy?

Response 1:

Thanks to the reviewer’s comment. The function of RANSAC is to suppress the noise influence and solve the problem of low SNR of the chessboard. The description in the manuscript is not appropriate, we change the sentence “by using local RANSAC, the sub-pixel coordinates of the projector with high accuracy can be obtained” to “by using local RANSAC, the problem of low SNR of the chessboard is solved, thus obtaining the sub-pixel coordinates of the projector” in revised manuscript (Line 79-80).

Question 2:

Line 80: In addition, instead of a lookup table, the mapping relationship between the absolute phase and the projector coordinates by the cubic polynomial can become more flexible for rapid 3D reconstruction. Actually, look up tables are tools to replace runtime computation with a simple array indexing. How using a fitting process instead of LUT makes the algorithm more flexible?

Response 2:

Thanks to the reviewer’s comment. 1) First, as the reviewer said, “look up tables are tools to replace runtime computation with a simple array indexing.” However, in our method, LUT will round up the phase values which may result in a loss of measurement accuracy. Moreover, there is little difference between LUT method and polynomial method in 3D reconstruction speed. So we choose to use the cubic polynomial mapping method. 2) The method mentioned in reference 17 uses LUT to obtain the parameters of phase to three-dimensional coordinates at each pixel position, since this method is based on additional sampling calibration. Therefore, the measurement accuracy outside the sampling range cannot be guaranteed. 3) Our method requires only a few numbers of polynomial parameters,not only guarantees the speed, but also has higher accuracy outside the calibration range. Moreover, it occupies less memory and the calibration process is simple than the method mentioned in reference 17.

In order to avoid ambiguity, we change the sentence “However, this method requires significant computation and needs additional calibration experiments to establish the mapping relationship after projector calibration, which increases the complexity and difficulty of practical application.” to “However, this method needs additional sampling calibration to establish the relationship between phase and coordinate, which can’t guarantee the measurement accuracy outside the sampling range, and the calibration results occupy a lot of system memory, thus limits the practical application.” in revised manuscript (Line 72-76).

Change the sentence “In addition, instead of a lookup table, the mapping relationship between the absolute phase and the projector coordinates by the cubic polynomial can become more flexible for rapid 3D reconstruction.” to “In addition, we establish the mapping relationship between the absolute phase and the projector coordinates by the cubic polynomial for more flexible rapid 3D reconstruction.” in revised manuscript (Line 81-83).

Question 3:

The proposed method should be compared to some baseline algorithms.

Response 3:

Thanks to the reviewer for reminding. We added a comparison experiment and compared the calibration reprojection errors with different methods in revised manuscript (Line 300). And we added a sentence “And the reprojection errors of the camera, projector, and both (the overall mean error) are calculated and compared in Table 2.” (Line 294-295)

Method

Reprojection errors (pixel)

Camera

projector

Overall mean error

Moreno & Taubin [18]

0.15

2.58

1.83

Global homography

0.15

7.45

5.27

Huang’s method [25]

0.26

0.17

0.22

Proposed

0.02

0.06

0.03

 Table 2. Calibration reprojection errors (pixels) with different methods

Question 4:

The English should be improved.

Response 4:

Thanks to reviewer’s comment. We have carefully reviewed and revised the paper and made the following modifications.

1) Line 43: Change the word “calculate” to “calculating”.

2) Line 47: Add the word “and”.

3) Line 51: Add the word “method”.

4)Line 117: Change the word “principle” to “principal”.

5) Line 165: Add the word “and”.

6) Line 199: Change the word “superscript” to “subscript”.

7) Line 212: Change the word “accordant” to “corresponding”.

8) Line 275: Change the phrase “10 × 10 mm2” to “10 × 10 mm2”.

9) Line 315: Change the phrase “time consuming” to “time-consuming”.

10) Line 320: Change the phrase “time consuming” to “time-consuming”.

11) Line 335: Change the word “disminishing” to “diminishing”.

Round 2

Reviewer 2 Report

I have no more comments. I recommend to accept the manuscript.